# Comprehensive quality assessment for aphasia rehabilitation after stroke: protocol for a multicentre, mixed-methods study

Sam Harvey [1,2] Marissa Stone [1,3] Sally Zingelman [1,2]
David A Copland [1,2] Monique F Kilkenny [4,5] Erin Godecke [6,7]
Dominique A Cadilhac [4,5] Joosup Kim [5,8] Muideen T Olaiya [4]
Miranda L Rose [7,9] Caterina Breitenstein [10] Kirstine Shrubsole [1,11]
Robyn O'Halloran [7,9] Annie J Hill [7,9] Deborah Hersh [12,13]
Kathryn Mainstone,[1] Penelope Mainstone,[1] Carolyn A Unsworth [4,14]
Emily Brogan [15,16] Kylie J Short,[2] Clare L Burns [17] Caroline Baker [7,18]
Sarah J Wallace [1,2]

SH, MS and SZ are joint first authors.

For numbered affiliations see end of article.

**Correspondence to**
Dr Sam Harvey;
sam.harvey@uq.edu.au

## ABSTRACT

**Introduction** People with aphasia following stroke experience disproportionally poor outcomes, yet there is no comprehensive approach to measuring the quality of aphasia services. The Meaningful Evaluation of Aphasia SeRvicES (MEASuRES) minimum dataset was developed in partnership with people with lived experience of aphasia, clinicians and researchers to address this gap. It comprises sociodemographic characteristics, quality indicators, treatment descriptors and outcome measurement instruments. We present a protocol to pilot the MEASuRES minimum dataset in clinical practice, describe the factors that hinder or support implementation and determine meaningful thresholds of clinical change for core outcome measurement instruments.

**Methods and analysis** This research aims to deliver a comprehensive quality assessment toolkit for poststroke aphasia services in four studies. A multicentre pilot study (study 1) will test the administration of the MEASuRES minimum dataset within five Australian health services. An embedded mixed-methods process evaluation (study 2) will evaluate the performance of the minimum dataset and explore its clinical applicability. A consensus study (study 3) will establish consumer-informed thresholds of meaningful change on core aphasia outcome constructs, which will then be used to establish minimal important change values for corresponding core outcome measurement instruments (study 4).

**Ethics and dissemination** Studies 1 and 2 have been registered with the Australian and New Zealand Clinical Trial Registry (ACTRN12623001313628). Ethics approval has been obtained from the Royal Brisbane and Women's Hospital (HREC/2023/MNHB/95293) and The University of Queensland (2022/HE001946 and 2023/HE001175). Study findings will be disseminated through peer-reviewed publications, conference presentations and engagement with relevant stakeholders including healthcare providers, policy-makers, stroke and rehabilitation audit and clinical quality registry custodians, consumer support organisations, and individuals with aphasia and their families.

## STRENGTHS AND LIMITATIONS OF THIS STUDY

⇒ This research will deliver a comprehensive, fit-for-purpose, quality assessment toolkit for aphasia care after stroke, ready for implementation in the Australian context.
⇒ This is a world-first pilot of a coproduced minimum dataset in poststroke aphasia services.
⇒ Our research will establish clinically meaningful benchmarks of change for core aphasia outcome measurement instruments in partnership with key stakeholders including people with lived experience of aphasia who will be involved in all phases of the research.
⇒ This pilot study is limited to the Australian metropolitan public healthcare context, which may limit translation to other geographical and care settings.
⇒ Future studies employing the Meaningful Evaluation of Aphasia SeRvicES minimum dataset could integrate behavioural and imaging data to gain a fuller understanding of factors influencing aphasia recovery.

## INTRODUCTION

Globally, 12.2 million people experience a stroke annually.[1] Up to 40% of stroke survivors initially experience aphasia,[2] which affects the ability to produce and understand language. Language underpins family, social and vocational interactions and accordingly, aphasia is associated with significant psychosocial impacts[3] and poor health-related quality of life.[4] Speech and language therapy

for aphasia is effective[5–7] and meta-analysis of large individual patient datasets reveals that people with aphasia who receive intensive, frequent, and tailored speech and language therapy achieve better outcomes.[8] In clinical practice, however, people with aphasia receive limited therapy[9–11] and finite clinical resources may not be used efficiently due to a lack of understanding of which treatments work best and for whom.[12 13] The delivery of high-value, personalised aphasia treatment that makes the best use of available resources requires an understanding of the characteristics of people with aphasia (sociodemographic factors, stroke type or severity and aphasia characteristics), the type and quality of care they receive, and the outcomes they desire and achieve. Currently, data to support understanding of aphasia care and outcomes are not routinely collected[14 15] and there are few benchmarks of patient-perceived therapy success to indicate when meaningful change has occurred.[16]

Health service monitoring is essential to the delivery of efficient, high-quality and safe healthcare.[17] Minimum datasets (standardised sets of variables) and core outcome sets (COS) (agreed outcomes and measurement instruments) are often collected within health services to enable benchmarking and drive quality improvement.[18] At an individual patient level, the routine collection of valid and reliable data can inform treatment decision-making and support the provision of personalised care.[19] At a service and/or population level, analysis of routinely collected data can provide insights into trends in quality of care and patient outcomes.[19] Consequently, routinely collected data can inform targeted quality improvement initiatives to reduce variation in patient care, increase alignment with clinical practice guidelines, optimise patient outcomes and support the development of policy to address the needs of patients and communities.[19–21]

In Australia, three national systems for stroke service monitoring exist, however, few aphasia-specific data variables are collected. The Australian Clinical Stroke Registry (AuSCR) aims to monitor and improve the quality of acute stroke care,[22] collecting patient demographics, type of stroke and clinical indicators of prioritised stroke management (eg, receipt of stroke unit care and swallowing assessment[23]) on consecutive admissions to participating hospitals. While the AuSCR is driving improvements in stroke care in Australia,[24] the only aphasia-specific information collected is the International Classification of Diseases code for aphasia.[25] In addition to the AuSCR, the Stroke Foundation National Audit Programme monitors and measures acute and rehabilitation care against the Australian Clinical Guidelines for Stroke Management, using a retrospective audit to inform quality improvement and advocacy.[22] Currently, only three quality measures relevant to aphasia are collected in the audit: the identification of speech/communication deficits and aphasia; assessment by a speech pathologist; and, identification of management strategies for those identified to have aphasia.[26] Finally, the Australasian Rehabilitation Outcomes Centre (AROC) dataset aims to drive quality and outcome improvements in rehabilitation through the collection of inpatient rehabilitation data. AROC captures the number of minutes of speech or language therapy provided, but there are no aphasia-specific variables included in this dataset.[27] A consistent and comprehensive approach to aphasia service monitoring is needed to drive targeted improvement in quality of care and patient outcomes.

## A comprehensive quality assessment toolkit for aphasia care after stroke: the MEASuRES minimum dataset

The MEASuRES (Meaningful Evaluation of Aphasia SeRvicES) minimum dataset was developed in partnership with researchers, clinicians and people with aphasia and their families, to address the gap in routine aphasia-specific data collection and reporting. MEASuRES (box 1) comprises four elements (sociodemographic characteristics, quality indicators, treatment descriptors, outcome measurement instruments) and builds on three international multistakeholder initiatives to standardise aphasia data and outcome collection and description: the DESCRIBE reporting standards project (DESCRIBE[28]), the Research Outcome Measurement in Aphasia COS (ROMA COS[29]) and the International Population Registry for AphasIa after StrokE (I-PRAISE[30]).

### Sociodemographic characteristics

People with aphasia are a heterogeneous population, which presents challenges in prognostication, treatment prescription and implementation of research evidence within clinical contexts. Factors such as age, sex, stroke severity, time poststroke onset, lesion site and size, aphasia type and severity, and comorbidities are likely to impact recovery and treatment outcomes.[5 31] Social determinants of health (eg, education) are hypothesised to impact aphasia outcomes, however, research in this area is limited[32] and these factors are generally under-reported in aphasia research.[28] The DESCRIBE project established internationally agreed minimum reporting standards for participant characteristics in aphasia research, in partnership with multidisciplinary researchers, clinicians and journal editors.[28]

### Quality indicators of clinical processes of care

Clinical guidelines present recommendations for important processes of care (eg, assessment, therapy, education). Guideline adherence can serve as indicators of the quality of care provided (ie, quality indicators).[33] Associations between adherence to processes of care and improved outcomes have been documented for acute stroke care[34] and stroke rehabilitation.[35] However, there is currently no systematic means of determining the quality of aphasia care. A set of quality indicators for poststroke aphasia services has been developed in partnership with clinicians, researchers and people with aphasia and their significant others.[36] These quality indicators are based on strong research evidence and processes of care identified as being important to stakeholders.

## Box 1   Data items included in the measures minimum dataset

### Sociodemographic characteristics[28]

**Person with aphasia.**
⇒ Age.
⇒ Biological sex.
⇒ Years of education.
⇒ Language of assessment and treatment.
⇒ Primary language spoken at home.
⇒ Languages used.
⇒ History of condition known to impact communication or cognition.
⇒ History of previous stroke.
⇒ Lesion hemisphere.
⇒ Time since onset of aphasia.
⇒ Conditions arising from the neurological event.

**Communication partner.**
⇒ Age.
⇒ Biological sex.
⇒ Relationship to person with aphasia.

### Quality indicators[36]
⇒ A screener and/or assessment is completed to determine if communication impairment (including aphasia) is present.
⇒ A valid and reliable standardised assessment is conducted to determine aphasia severity.
⇒ Information about aphasia is provided to the person with aphasia.
⇒ Information about aphasia is provided to the person with aphasia's primary carer/communication partner.
⇒ Information about support is provided to the person with aphasia's primary carer/communication partner.
⇒ The primary carer/communication partner of the person with aphasia is provided with communication partner training.
⇒ Individualised recommendations for communicating with the person with aphasia are provided to the treating team.
⇒ There is training for staff in supported communication for aphasia.
⇒ Goal setting is undertaken in partnership with the person with aphasia and their significant others.
⇒ The person with aphasia receives person-centred/family-centred care.
⇒ The person with aphasia receives speech and language therapy.

### Treatment descriptors[30]
⇒ Treatment setting.
⇒ Treatment target.
⇒ Therapeutic approach.
⇒ Treatment type.
⇒ Number and duration of treatment sessions provided.
⇒ Delivery mode.

### Outcomes

**Research Outcome Measures in Aphasia Core Outcome Set.[29 68]**
⇒ The Western Aphasia Battery-Revised (Language).
⇒ The Scenario Test (communication).
⇒ Stroke and Aphasia Quality of Life Scale-39g (QoL).
⇒ The General Health Questionnaire-12 (emotional well-being).

**Minimal important change.[44]**
⇒ Anchor ratings for each construct (language, communication, QoL, emotional well-being).

## Treatment descriptors

The I-PRAISE[30] includes a protocol for the documentation of aphasia treatment delivery. Information regarding treatment types, methods of delivery and treatment dose are required to inform an understanding of poststroke aphasia treatments.[12 37 38] These treatment descriptors would support the identification of outcome predictors,[12] however, they are not routinely collected.

## Core outcome measurement instruments

The outcomes achieved by people with aphasia after treatment are defined by change scores on standardised measurement instruments. The ROMA COS defines an internationally agreed standardised protocol for measuring treatment outcomes. The four measurement instruments included in the ROMA COS—Western Aphasia Battery: Revised,[39] The Scenario Test,[40] Stroke and Aphasia Quality of Life Scale-39g (SAQOL-39g)[41] and General Health Questionnaire-12[42]—map directly to constructs that people with aphasia, their families, clinicians and researchers have identified as being essential to understanding aphasia and its consequences (ie, language, communication, quality of life, emotional well-being).[43] The ROMA COS is designed for use in aphasia research and the feasibility and acceptability of its routine use in clinical settings requires evaluation.

## Interpretation of treatment success: minimal important change

Treatment success cannot be determined by change on standardised outcome measurement instruments alone.[16] To understand if aphasia treatments are successful from the patients' perspective, there is a need for objective benchmarks which quantify clinically meaningful changes.[12 16] These perspectives, alongside outcome measurement change scores for the individual patient, can be used to establish minimal important change (MIC) values.[44] MIC is the smallest change in the outcome measurement score, in the construct to be measured, which patients perceive to be important.([45]p. 408) MIC values have been proposed as a method of interpreting outcomes from the patient perspective and may be an indicator of whether treatment was successful, should continue, or be stopped, delayed, or changed.[46] MIC bridges the gap between statistical significance of an individual test score and its clinical importance and will facilitate interpretation of individual outcome measurement scores, supporting clinicians to provide high-quality person-centred care,[16] as well as allowing comparison of treatment responder rates across healthcare facilities.

There are few existing MIC values for aphasia measurement instruments. Guo *et al*[47] conducted an estimation study with 78 stroke survivors with and without aphasia, and included both the English and Mandarin versions of the SAQOL-39g. The SAQOL-39g (included in our minimum dataset) was completed by 34.6% of the participants (n=27). One-point improvement on the Modified Rankin Scale,[48] a measure of disability rather than quality of life, was used as the threshold of clinically meaningful change

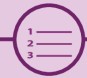

## MEASUREMENT TOOL
### What to measure in aphasia clinical care

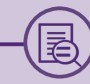

**STUDY 1**

Pilot the MEASuRES minimum dataset

———————

Multi-centre observational study

———————

200 people with aphasia 5 health services

———————

Data collection January – August 2024

**STUDY 2**

Evaluate the process of piloting the dataset

———————

Mixed-methods process evaluation

———————

Study 1 participants & health service clinicians

———————

Data collection July – August 2024

These studies will evaluate the use of the MEASuRES minimum dataset in clinical practice

## ANALYSIS TOOL
### How to interpret treatment success

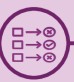

**STUDY 3**

Determine thresholds of meaningful change

———————

Focus groups & consensus workshop

———————

10 people with aphasia 10 clinicians

———————

Data collection April – May 2023

**STUDY 4**

Establish Minimal Important Change values

———————

Integration of aphasia outcomes and patient perspectives from Study 1 & Study 3

———————

Analysis to be conducted at the completion of Study 1 & Study 3 data collection

These studies will establish Minimal Important Change values for core aphasia outcome measurement instruments

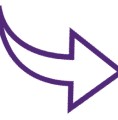 COMPREHENSIVE QUALITY ASSESSMENT PROGRAM FOR APHASIA CARE AFTER STROKE 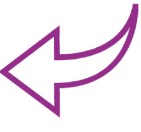

**Figure 1** Four linked studies will deliver a comprehensive quality assessment programme for aphasia care after stroke. < Figure 1 Four linked studies will deliver a comprehensive quality assessment programme for aphasia care after stroke >

to establish the MIC values. There are no known MIC values for the other outcome measurement instruments of the MEASuRES minimum dataset.[16]

## AIMS

Our objective is to deliver a comprehensive quality assessment toolkit for aphasia care after stroke. Specifically, we aim to:
1. Evaluate the use of the MEASuRES minimum dataset in clinical practice (studies 1 and 2).
2. Establish MIC values for core aphasia outcome measurement instruments, informed by stakeholder perspectives of meaningful change (studies 3 and 4).

## METHODS AND ANALYSES

A comprehensive quality assessment toolkit for aphasia care after stroke will be delivered in four studies (see figure 1 for overview). Studies 1, 2 and 4 will involve speech pathology clinicians collecting components of the MEASuRES minimum dataset and patient perceptions of change alongside routine hospital care. These data will be collected for 200 participants with aphasia throughout the continuum of care across participating health service sites between January and August 2024. The sample size was informed by recommendations for conducting and reporting MIC studies,[46] aphasia admission rates at each of the participating sites, and power calculations for subgroup analyses for studies 1 and 4. In conjunction with the multicentre study, an expert panel of stakeholders will establish a threshold of meaningful change in study 3. Data collection for study 3 occurred across April and May 2023. Recruitment processes and data analysis for each study are detailed below. Together, the four studies will provide preliminary information about the acceptability and feasibility of using the MEASuRES minimum dataset in routine clinical care. Embedded processes will allow exploration of where, when, and why components of the minimum dataset are not or cannot be collected to inform future refinement and scaling of the minimum dataset.

### Patient and public involvement

Two consumer investigators (KM and PM) and a research advisory group are involved with the project. The consumer investigators with lived experience of aphasia

provide expert guidance, feedback and oversight across the project, including contributions to the project design, funding applications and development of the research protocol. The research advisory group comprises people with aphasia (n=3), family members (n=2) and speech pathologists (n=2). The advisory group has contributed to recruitment strategies, study procedures and participant-facing materials. Contributions will continue through interpretation and codissemination of findings through peer-reviewed journal articles and conference presentations. Research advisory group members are remunerated, commensurate with Health Consumers Queensland guidelines,[49] to convene quarterly, and as needed, to complete specific tasks. Consumer and community involvement will be reported in line with the Guidance for Reporting Involvement of Patients and the Public(GRIPP2) long-form checklist[50] in future publications.

### Development of the measurement tool

The measurement tool will be used to collect standardised information about people with aphasia, aphasia care and aphasia outcomes in routine clinical care. The development of the measurement tool involves piloting the MEASuRES minimum dataset in clinical practice (study 1) and an embedded process evaluation (study 2).

### Study 1: piloting the MEASuRES minimum dataset

The aim of this study is to administer the MEASuRES minimum dataset in clinical settings to describe participant characteristics, and the delivery and outcomes of poststroke aphasia services. The minimum dataset will be administered by speech pathologists alongside routine clinical care.

### Study design and setting

This multicentre observational pilot study will be conducted in a convenience sample of five Australian publicly funded, metropolitan, tertiary health services providing stroke care: Metro North Hospital and Health Service (Queensland), St Vincent's Hospital Melbourne (Victoria), Monash Health (Victoria), Sir Charles Gairdner Osborne Park Health Care Group (Western Australia) and Fiona Stanley and Fremantle Hospital Group (Western Australia).

### Participants and sample size

People with aphasia (n=200) who meet the following eligibility criteria will be invited to participate: (1) admitted to a participating health service site; (2) new onset of aphasia caused by a new acute stroke, (3) aged ≥18 years and (4) has capacity to consent by self or with a proxy.

### Recruitment and consent

Patients admitted to participating health service sites with a new acute stroke will be screened for eligibility by clinical speech pathologists. Stroke diagnosis will be determined by the treating stroke physician. Eligible participants will be provided with written, aphasia-friendly participant

information and consent forms describing the project, its aims and the research activities. Speech pathologists will provide communication support throughout the consenting process.

Capacity to consent: Speech pathologists working within the health service sites will determine a potential participant's capacity to consent. Following a discussion about the study using appropriate supported communication techniques, prospective participants will be asked a series of yes/no questions about the study to establish if they have understood the information presented. Patients who answer all questions correctly will be considered to have capacity to consent.

Consent via proxy: If a prospective participant agrees to participate but is not able to demonstrate capacity to provide informed consent at the time of recruitment, a proxy will be invited to consent on their behalf (Queensland and Victorian sites only). In such cases, treating clinicians will continue to monitor the participants' recovery and will provide further opportunities for the participant to provide their own consent. Patients admitted to Western Australian sites who are unable to demonstrate capacity to consent will be ineligible to participate, in accordance with Western Australian Department of Health legislation.

### Data collection and management

The minimum dataset variables will be collected by speech pathologists using a purpose-built REDCap database.[51] The continuity of speech pathologists collecting data will be dependent on usual care practices and service structure at each health service site. There will be a minimum interval of two weeks between administrations of aphasia outcome measurement instruments, which is longer than the minimum test–retest reliability assessment interval for most measurement instruments.[41 52] In instances where elements of the minimum dataset are unable to be collected, reasons for missing or incomplete data will be recorded. Routine data validation will be undertaken to monitor completeness of data and identify invalid data entries. Data quality and completeness will be enhanced by using the logic checks, mandatory fields and duplicate report flagging tools.

### Planned analyses

Data quality and completeness will be summarised using descriptive statistics. For each variable in the minimum dataset, the proportion completed will be calculated to determine patterns of missing, incomplete or invalid data. Missing data will be handled in accordance with the processes outlined in existing guidelines.[53 54]

Similarly, descriptive statistics will be used to summarise patient characteristics, adherence to quality indicators, treatment descriptors and scores on aphasia outcome measurement instruments. Adherence to quality indicators will be defined as the proportion of eligible episodes of stroke in which the indicator was provided, estimated overall and at the hospital level. We will also summarise

the total number of indicators provided per episode at each hospital. Associations between patient characteristics, adherence to quality indicators and aphasia outcomes will be determined using multilevel regression models, adjusted for clustering effect within the hospital. The type of regression models to be used for analysis will depend on the nature and distribution of the outcome variables. All regression models will also be adjusted for covariates, including age and baseline scores on the aphasia outcomes and other covariates based on data collected.[53]

### Study 2: process evaluation of piloting the MEASuRES minimum dataset

This process evaluation will evaluate the performance of the MEASuRES minimum dataset, and to explore its acceptability and feasibility in clinical settings, from the perspective of patients and speech pathologists.

### Study design and setting

This mixed-methods process evaluation[55] will involve (a) patient and clinician surveys completed during administration of the minimum dataset (study 1); (b) semistructured interviews conducted with speech pathology clinicians following completion of the pilot study (study 1) and (c) integration of REDCap and site-specific training and implementation data.

### Recruitment and consent

The process evaluation will involve two participant groups: people with aphasia and speech pathologists. All people with aphasia (or their proxy) who consented to participate in study 1 (maximum n=200) will be invited to complete a survey prior to discharge from hospital to explore their experiences of completing the minimum dataset. When consenting for study 1, participants will be provided with written, aphasia-friendly participant information and consent forms describing the research activities associated with study 2. Speech pathologists will provide communication support throughout the consenting process. The survey will be in an aphasia-friendly format and administered in-person by speech pathology clinicians with provision of communication support.

All speech pathologists who administer the minimum dataset data in study 1 (n= ~25) will be invited to complete two online surveys throughout (time point 1) and at the completion (time point 2) of study 1, to explore the performance of the minimum dataset, and their perspectives of administering the minimum dataset in clinical settings. In addition, expressions of interest will be sought from speech pathologists to participate in semistructured interviews at the completion of study 1. Of those who express interest, 15 clinicians who meet the following eligibility criteria will be invited to participate in the interviews: (1) being a qualified speech pathologist, (2) being employed by a participating health service and (3) having participated in the pilot administration of the MEASuRES minimum dataset (study 1). Purposive sampling of clinicians across years of experience, clinical

setting and health service sites will be applied to ensure a range of viewpoints. These interviews will serve to contextualise survey results and provide further insights from the participating clinicians. Additional interviews will be conducted where required. Consent will occur in-person or online, following the opportunity to discuss participation with the research team. The consent process will be conducted by a staff member of the research team.

### Data collection

Survey elements will be designed in alignment with previous surveys used to evaluate COS and minimum dataset implementation.[56] Semistructured interview questions will be based on previous research investigating barriers and facilitators to COS use and aphasia treatment delivery,[57] and will consider all aspects of piloting the minimum dataset (ie, including collection of demographic data and quality indicators). Acceptability and feasibility will be explored through survey and semistructured interview data. A log of site-training and factors influencing implementation of the minimum dataset will be maintained, along with routinely collected administrative data.

### Planned analyses

Descriptive statistics will be used to summarise quantitative data from the surveys and interviews.[53] Qualitative survey and interview data will be analysed using content analysis,[58] to explore the performance of the minimum dataset and to understand the experiences of people with aphasia and speech pathology clinicians. The qualitative data will be compared and integrated with quantitative findings using a convergent interactive design.[59] Barriers and facilitators to administering the dataset will be mapped to the theoretical domains framework[60] and behaviour change wheel[61] to support future scaling.

Findings from the different data sources (surveys, interviews, REDCap, training and implementation logs) will be integrated and compared using the Medical Research Council process evaluation framework[55] to understand the relationships between implementation, contextual factors and other mechanisms of impact.

### Development of the analysis tool

The analysis tool is designed to provide objective interpretation of meaningful changes following aphasia treatment. In study 3, we will determine thresholds of meaningful change on patient-reported anchor scales (figure 2). These anchor scales were adapted from the anchor question proposed by Revicki et al to be conceptually relevant to core outcomes in aphasia (ie, language, communication, quality of life and emotional well-being[62]). As a patient-rated measure, labels for the indicators on the anchor scale were selected based on ease of understanding by patients with aphasia. While an indicator for +3 (completely recovered) represents a potential perceived outcome for participants early after stroke onset, an acceptable indicator for −3 (a point lower

## How much has your language changed since you last did this test?

| Much Worse | Slightly Worse | No Change | Slightly Improved | Much Improved | Completely Recovered |
|---|---|---|---|---|---|
| 👎👎 | 👎 | 😐 | 👍 | 👍👍 | 👍👍👍 |
| | | | | | |

**Figure 2** Example of a patient-reported 6-point Likert anchor scale, adapted from previous research.[62]

than −2=much worse) could not be identified. As such, the anchor scale was developed with six response categories including a neutral response option, in line with existing research.[63] In study 4, the agreed thresholds on the anchor scale (study 3) will be applied to treatment outcome data (study 1), establishing MIC values for the ROMA COS.

### Study 3: establishing consensus-based thresholds for the calculation of MIC values

The aims of study 3 are to determine thresholds of meaningful change from the perspectives of people with aphasia and speech pathologists. Stakeholder perspectives have seldom been included in determining the threshold of clinically meaningful change on anchor rating scales.[64] By undertaking a stakeholder-focused approach, the resulting MIC values will be clinically meaningful and relevant to the experiences of people with aphasia.[65 66] Outcomes from this study will inform the anchor-based predictive modelling methodology for establishing MIC values (see study 4).

### Setting and design

People with aphasia and speech pathologists will participate in online focus groups and a consensus meeting to discuss clinically important change in aphasia, and to establish the threshold for a meaningful change based on the anchor scales (figure 2).

### Participants and sample size

An expert panel of people with poststroke aphasia (n=10) and speech pathologists (n=10) will be convened. Exclusion criteria for people with aphasia include: (a) aetiology other than stroke, (b) neurological, cognitive or sensory condition that impedes ability to participate and (c) being unable to verbally or non-verbally communicate a yes/no response. Exclusion criteria for speech pathologists: does

not work in poststroke aphasia rehabilitation. People with aphasia will be purposively sampled for: sex (male, female), time poststroke (1 week to 6 months, >6 months); age (<65 years, ≥65 years of age) and severity of aphasia (mild, moderate, severe aphasia per the Aphasia Severity Rating Scale[67]). Speech pathologists will be purposively sampled for work setting (inpatient rehabilitation, community rehabilitation). Participants will be consented to the study in alignment with the sampling criteria. It is anticipated that 10 people with poststroke aphasia and 10 speech pathologists will be sufficient to inform a threshold for meaningful change. Each participant will attend one focus group. A maximum of five participants will attend each per group. The final number of focus groups will be determined in part by the amount of time and level of communication assistance each participant will require to understand and meaningfully participate, and in part by the analysis of focus group findings. All participants will attend the consensus workshop.

### Recruitment and consent

Expressions of interest will be sought from people with aphasia, via local and national stroke and aphasia research centres and consumer organisations including their social media channels. All prospective participants will be provided with communicatively accessible information about the study. A meeting will be held to answer questions and obtain demographic details for sampling. Participants who respond correctly to yes/no questions about the study's purpose and participation requirements will be invited to consent. Project investigators who are speech pathologists will provide communication support throughout the consenting process.

To reduce the burden of participation, honorarium payments will be offered for time spent participating in study 3. Participants with aphasia will be remunerated

at a rate of $A40 per hour, for an estimated total of five hours per participant (consent meeting, focus group, consensus workshop), in accordance with recommendations from Health Consumers Queensland.[49] Speech Pathologists will be reimbursed at a rate of $A60 per hour for an estimated total of four hours per participant (focus group, consensus workshop), commensurate with the Queensland Health HP4 casual hourly rate.

### Data collection and planned analyses

Focus groups and the consensus workshop will be recorded and transcribed. Data will be analysed using content analysis.[58] In the consensus workshop, participants will vote on whether different thresholds of meaningful change should be established for participant subgroups (eg, aphasia severity or time post-onset). Participants will vote on the threshold for important change on anchor scales for each construct (ie, language, communication, emotional well-being, quality of life) as these may differ. Quantitative data will be analysed using descriptive statistics (percentages of responses). Consensus will be defined a priori as ≥70% agreement.

### Study 4: establishing MIC values for the ROMA COS

This study will involve integrating and analysing data from studies 1 and 3 to establish MIC values for the ROMA COS.[44]

### Design

Anchor-based predictive modelling approach to establishing MIC.

### Data collection

As part of study 1, the ROMA COS will be administered prior to and after usual care aphasia intervention, with a minimum interval of 2 weeks. When consenting for study 1, participants will be provided with written, aphasia-friendly participant information and consent forms describing the research activities associated with study 4. Speech pathologists will provide communication support throughout the consenting process. With the second administration of the ROMA COS, both participant groups (people with aphasia and clinicians) will additionally rate perceived change in language, communication, quality of life and emotional well-being on corresponding anchor scales (figure 2). The second ROMA COS and anchor scales will be collected prior to discharge from inpatient care at the participating health service site. In instances where this is not feasible, outpatient follow-up will be undertaken, following a flexible approach to delivery mode and assessor in line with site-specific agreements.

### Planned analyses

MIC values will be estimated for each of the ROMA COS instruments using an anchor-based predictive modelling approach.[44] Participants' responses on the anchor rating scale will be dichotomised as either 'improved' or 'not improved' based on the thresholds of meaningful improvement established in study 3. Likelihood ratios will then be calculated with the change scores using logistic regression models, following the methods described by Terluin *et al*.[44] Logistic regression models will comprise the nominal anchor response (improved/not improved) as the dependent variable, and the change score of each of the ROMA COS instruments from before to after the intervention as the independent variable.[44 46] The resulting MIC values for each of the ROMA COS instruments will indicate the smallest change from baseline which is perceived as important by patients with aphasia and allows classification of treatment success in treatment responders and non-responders.

### Anticipated outcomes

Collectively, findings from these four studies will deliver a comprehensive quality assessment toolkit for aphasia care after stroke. Findings from the pilot study (study 1) will provide a detailed understanding of the characteristics of people with aphasia, and their aphasia care and treatment-induced outcomes. The process evaluation (study 2) will contribute a comprehensive understanding of the performance of the MEASuRES minimum dataset, including feasibility and acceptability from the perspectives of speech pathologists and people with aphasia. The combined findings of these two studies will inform the development of a measurement tool for aphasia services and support future scaled implementation. Study 3 will establish thresholds of meaningful change on patient-reported anchor scales for core aphasia constructs (ie, language, communication, emotional well-being, quality of life) and various subgroups of the aphasia population, which will be used to determine MIC values for the ROMA COS in study 4. Study 3 and study 4, combined, will contribute to the development of an analysis tool for interpreting aphasia outcomes at the individual level. Beyond these immediate outcomes, MEASuRES data items may be considered for inclusion in existing stroke research datasets and clinical registries. Site-specific information regarding potential evidence-practice gaps may drive future quality improvement activities.

### Ethical considerations

Approval was obtained for studies 1, 2 and 4 from the Queensland Health Metro North Hospital and Health Service Human Research Ethics Committee B (approval ID HREC/2023/MNHB/95293) and for study 3 from The University of Queensland Human Research Ethics Committee (approval ID 2022/HE001946). Studies 1 and 2 have been registered with the Australian and New Zealand Clinical Trial Registry (ACTRN12623001313628). Site-specific governance approvals are being sought at time of publication.

### Risks

Participation is unlikely to result in any significant harm. However, people with aphasia risk experiencing grief and distress when asked to reflect on their experiences during hospitalisation with a new stroke and aphasia. Individuals

with aphasia may be more susceptible than the general stroke population to discomfort or psychological distress as a result of their communication difficulties. A Distress Protocol has been developed to guide clinicians in managing and mitigating participant distress.

## Dissemination plan

We intend to report findings of this project in international peer-reviewed journal articles and at relevant national and international conferences. A summary of research findings will be made available to participating sites. Results will be communicated to clinicians through local, state-wide, national and international clinical and research networks and forums. Communication accessible summaries of this research will be made available for people with aphasia. Research advisory group members with lived experience of aphasia, alongside project investigators, will seek opportunities to present at aphasia-specific forums such as the Australian Aphasia Association national conference. Multimodal research summaries will be reviewed by the research advisory group and shared across platforms supporting aphasia-friendly communication, including the Queensland Aphasia Research Centre's Matters mailing list, the Centre of Research Excellence in Aphasia Recovery and Rehabilitation Community of Practice, the Australian Aphasia Association social media pages and Aphasia Recovery Connection.

## Progress update

Data collection for study 3 has been completed and analysis is underway at the time of publication.

## Author affiliations

[1] Queensland Aphasia Research Centre, The University of Queensland, Saint Lucia, Queensland, Australia
[2] Surgical, Treatment and Rehabilitation Service Education and Research Alliance, The University of Queensland and Metro North Hospital and Health Service, Herston, Queensland, Australia
[3] St Vincent's Hospital Melbourne Pty Ltd, Fitzroy, Victoria, Australia
[4] Department of Medicine, School of Clinical Sciences at Monash Health, Monash University, Clayton, Victoria, Australia
[5] Stroke Theme, The Florey Institute of Neuroscience and Mental Health, Parkville, Victoria, Australia
[6] School of Medical and Health Sciences, Edith Cowan University, Joondalup, Western Australia, Australia
[7] Centre for Research Excellence in Aphasia Recovery and Rehabilitation, Melbourne, Victoria, Australia
[8] Department of Medicine, School of Clinical Sciences, Monash University, Clayton, Victoria, Australia
[9] School of Allied Health, Human Services and Sport, La Trobe University College of Science Health and Engineering, Bundoora, Victoria, Australia
[10] Department of Neurology with Institute of Translational Neurology, University of Muenster, Muenster, Germany
[11] Metro South Hospital and Health Service, Princess Alexandra Hospital, Woolloongabba, Queensland, Australia
[12] Curtin School of Allied Health and EnAble Institute, Curtin University, Perth, Western Australia, Australia
[13] Australian Aphasia Association, Perth, Western Australia, Australia
[14] Institute of Health and Wellbeing, Federation University, Ballarat, Victoria, Australia
[15] Edith Cowan University, Joondalup, Western Australia, Australia
[16] Fiona Stanley Fremantle Hospitals Group, South Metropolitan Health Service, Palmyra, Western Australia, Australia
[17] Royal Brisbane and Women's Hospital, Metro North Hospital and Health Service, Herston, Queensland, Australia
[18] Speech Pathology Department, Monash Health, Clayton, Victoria, Australia

**Acknowledgements** The authors acknowledge the support of the Collaboration of Aphasia Trialists which is funded by COST and The Tavistock Trust for Aphasia in fostering international aphasia research collaboration and in identifying benchmarks of meaningful change as a priority area for development.

**Contributors** Conceptualisation: SJW, CBreitenstein, DACopland, MFK, EG, DACadilhac, CU, KS, RO'H, MO, JK, AH, DH and MR. Protocol design and preparation. SJW, MS, SZ, SH, DACopland, MFK, EG, DACadilhac, CU, KS, RO'H, MO, JK, AH, DH, KM, PM, EB, KS, CBurns and CBaker. All authors had input on the submitted version of the manuscript.

**Funding** This project has been funded by the Medical Research Future Fund (project ID: MRF2016134, CIA SJW). SJW is funded by an NHMRC Investigator Grant (1175821). DACadilhac is supported by an NHMRC senior research fellowship (1154273). MFK is supported by a National Heart Foundation of Australia Future Leader fellowship (105737). MS and SZ are supported by Australian Government Research Training Program Scholarships.

**Competing interests** None declared.

**Patient and public involvement** Patients and/or the public were involved in the design, or conduct, or reporting, or dissemination plans of this research. Refer to the Methods section for further details.

**Patient consent for publication** Not applicable.

**Provenance and peer review** Not commissioned; externally peer reviewed.

**ORCID iDs**
Sam Harvey http://orcid.org/0000-0002-4839-2117
Marissa Stone http://orcid.org/0000-0001-6987-1822
Sally Zingelman http://orcid.org/0000-0003-0599-5708
David A Copland http://orcid.org/0000-0002-2257-4270
Monique F Kilkenny http://orcid.org/0000-0002-3375-287X
Erin Godecke http://orcid.org/0000-0002-7210-1295
Dominique A Cadilhac http://orcid.org/0000-0001-8162-682X
Joosup Kim http://orcid.org/0000-0002-4079-0428
Muideen T Olaiya http://orcid.org/0000-0002-4070-0533
Miranda L Rose http://orcid.org/0000-0002-8892-0965
Caterina Breitenstein http://orcid.org/0000-0002-6408-873X
Kirstine Shrubsole http://orcid.org/0000-0002-7805-2447
Robyn O'Halloran http://orcid.org/0000-0002-2772-2164
Annie J Hill http://orcid.org/0000-0003-3907-8369
Deborah Hersh http://orcid.org/0000-0003-2466-0225
Carolyn A Unsworth http://orcid.org/0000-0001-6430-2823
Emily Brogan http://orcid.org/0000-0001-9604-4558
Clare L Burns http://orcid.org/0000-0002-9752-1739
Caroline Baker http://orcid.org/0000-0001-8605-5181
Sarah J Wallace http://orcid.org/0000-0002-0600-9343

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
