## [Reviewer comments · BMJ Open]

ARTICLE DETAILS

TITLE (PROVISIONAL)	Comprehensive quality assessment for aphasia rehabilitation after stroke: Protocol for a multi-centre, mixed-methods study
AUTHORS	Harvey, Sam; Stone, Marissa; Zingelman, Sally; Copland, David; Kilkenny, Monique F; Godecke, Erin; Cadilhac, Dominique; Kim, Joosup; Olaiya, Muideen; Rose, Miranda; Breitenstein, Caterina; Shrubsole, Kirstine; O'Halloran, Robyn; Hill, Annie; Hersh, Deborah; Mainstone, Kathryn; Mainstone, Penelope; Unsworth, Carolyn; Brogan, Emily; Short, Kylie; Burns, Clare; Baker, Caroline; Wallace, Sarah J.

VERSION 1 – REVIEW

REVIEWER	Daniel Schindel Charité Universitätsmedizin Berlin, Institute of Medical Sociology and Rehabilitation Science
REVIEW RETURNED	31-Oct-2023

GENERAL COMMENTS	Review bmjopen-2023-080532 The project aims to establish a comprehensive and consistent framework for assessing aphasia services, with the goal of standardizing monitoring procedures and driving targeted enhancements in the quality of care. Consequently, this project represents a novel and clinically significant advancement in the realm of stroke aftercare. Below, I present my observations on the protocol. Page 9, Line 54 - MEASuRES: It's important to provide a clear definition or reference for MEASuRES. Including key items/domains in a table format might be a good idea for clarity. Page 10, Line 5 - Choice of Hospitals: Justifying the selection of hospitals and providing information about their structure, size, and representativeness for standard care in Australia might be essential for the study's external validity. Page 10, Line 3f - Planned Implementation Dates: Including information about the planned implementation dates for each study is important to understand the study's timeline and feasibility. Page 10, Line 14 - Sample Size Justification: Explaining the rationale behind the chosen sample size and the potential patient population in the participating clinics is necessary to assess the study's statistical power and generalizability. Page 10, Line 22 – Please clarify whether the diagnosis within the last year is a recruitment criterion or not? Do you differentiate between acute and chronic patients in the analysis? Is that important for you? Page 10, Line 26-27 - Providing Information for Studies 1, 2, and 4: Ensure that information is consistent for all studies, and it may be necessary to provide information for all studies conducted as part of the project, also study 3?
--

	Page 10, Line 37 – Have you discussed further ways to obtain informed consent? Especially for non-responsive patients. If people can't provide informed consent or are at the moment of recruitment non-responsive a proxy or legal representative might be invited to consent. Page 10, Line 44 - Patient Interviews: Clarify whether speech pathologists will interview the patients or if patients are expected to complete the questionnaire themselves (online/paper). Page 10-11 - Planned Analyses: Consider categorizing outcomes into primary and secondary outcomes for clarity in your analysis plan. Page 11, Line 16 - Missing Data: Since handling missing data is only need in time series, how many time points do you have? I can't find any information on how often the questionnaire is answered? What is the main observation duration? 1 week or 1 month after initial stroke? Please define. Page 12, Line 52 - Scale Balance: Ensure the scale is balanced in both directions to prevent bias, and reevaluate the necessity of the "completely recovered" category. Page 13, Line 15-30 - Well described. Patients and Proxy Involvement: Include information on whether participants received incentives, the type, and the amount (e.g., monetary compensation). Last note: The quantity of authors, particularly the abundance of first authors, appears notably high. Nonetheless a great project and endeavor. I look forward to the results of the studies.
--	---

REVIEWER	Deborah Levy University of California San Francisco, Neurosurgery
REVIEW RETURNED	18-Nov-2023

GENERAL COMMENTS	This study protocol is a well-written and well-motivated plan to measure the quality of aphasia services received following stroke using four related studies: (1) administration of a minimum dataset of assessments (MEASuRES) with acute stroke patients longitudinally across multiple clinical sites in Australia; (2) assessment of the acceptability and feasibility of acquiring MEASuRES in clinical settings; (3) establishment of focus groups with stakeholders to determine consensus on Minimal Important Change (MIC) metrics for aphasia; and (4) establishment of specific MIC values for differences in language, communication, QOL, and emotional wellbeing (as assessed within MEASuRES) following speech-language treatment. The protocol appears excellent overall, and I commend the authors for their focus on meaningful over statistically significant differences as well as their commitment to including stakeholders in the research process. Minor concerns:  - What is the specific plan for acquisition of longitudinal data (i.e., will the same speech pathologists from the acute setting reach out to these patients and schedule follow-up research appointments? Will these be conducted in person or remotely? With what frequency will these evaluations be acquired? What are the preferred assessment time points beyond > 2 weeks post-stroke? etc.) - While this will be investigated in Study 2, the feasibility of acquiring the full MEASuRES dataset multiple times within the tumultuous period immediately following stroke may be reasonably expected to be somewhat limited. Do the authors have a hypothesis of how likely this is to be achievable, and/or a backup assessment plan if it proves to be too difficult to achieve? This may warrant addition to
--

	the limitations section.  - The plan for disentangling recovery due to time / neuroplasticity compared to recovery due to treatment could be made clearer; is including time post stroke in regression models sufficient to address this? - Do the authors have a policy for or against incorporating neurological variables such as lesion size / lesion location into their models? These factors are likely to play a large role in the extent of recovery, and additionally may be confounded with treatment received. This may make interpretation of the specific influence of speech and language therapy in recovery more difficult. Do the authors have thoughts on the extent to which this is an issue for their purposes? - How will the authors determine what speech language treatment was received by each patient? Is the assumption that this will always occur within the same hospital system that acquired the MEASuRES dataset and therefore be easily trackable? - How were the desired sample sizes (e.g. 200 people with aphasia in Study 1, 10 each people with aphasia and speech pathologists in Study 3) chosen?
--	--

VERSION 1 – AUTHOR RESPONSE

Reviewer	Comment	Response	Changes
			Line numbers refer to the continuous line numbers on the marked version of the revised manuscript New text in red.
1	Page 9, Line 54 - MEASuRES: It's important to provide a clear definition or reference for MEASuRES. Including key items/domains in a table format might be a good idea for clarity.	Good idea. A new table (Table 1) lists the data items included in the MEASuRES minimum dataset.	Table 1 inserted in the Introduction section (line 97).
1	Page 10, Line 5 - Choice of Hospitals: Justifying the selection of hospitals and providing information about their structure, size, and representativeness for standard care in Australia might	A brief description of the participating health service characteristics has been added. Additionally, the section 'Strengths and limitations of this protocol' includes the following point which, we believe, speaks to	Lines 202-203 This multi-centre observational pilot study will be conducted in a convenience sample of five Australian publicly funded, metropolitan, tertiary health services providing stroke care:

	be essential for the study's external validity.	the issue of representativeness in the Australian context:  This pilot study is limited to the Australian metropolitan public healthcare context which may limit translation to other geographical and care settings. 	
1	Page 10, Line 3 - Planned Implementation Dates: Including information about the planned implementation dates for each study is important to understand the study's timeline and feasibility.		Lines 165-169 Studies 1, 2 and 4 will involve speech pathology clinicians collecting components of the MEASuRES minimum dataset and patient perceptions of change alongside routine hospital care. These data will be collected for 200 participants with aphasia throughout the continuum of care across participating health service sites between January and August 2024.
1	Page 10, Line 14 - Sample Size Justification: Explaining the rationale behind the chosen sample size and the potential patient population in the participating clinics is necessary to assess the study's statistical power and generalizability.	Thank you for this question. The following information has now been added in the 'Methods and Analyses' section. Justification of the sample size has also been added to Study 2 and Study 3.	Lines 167-171 These data will be collected for 200 participants with aphasia throughout the continuum of care across participating health service sites between January and August 2024. The sample size was informed by recommendations for conducting and reporting MIC studies [47], aphasia admission rates at each of the participating sites, and power calculations for sub-group analyses for Studies 1 and 4. Lines 285-287 Purposive sampling of clinicians across years of experience, clinical setting, and health service sites will be applied to ensure a range of viewpoints. These interviews will serve to contextualise survey results and provide further insights from the participating clinicians. Additional interviews will be conducted

			where required. Lines 351-357 Participants will be consented to the study in alignment with the sampling criteria. It is anticipated that 10 people with post-stroke aphasia and 10 speech pathologists will be sufficient to inform a threshold for meaningful change . Each participant will attend one focus group. A (maximum of five participants will attend each per group.). The final number of focus groups will be determined in part by the amount of time, and level of communication assistance, each participant will require to understand and meaningfully participate, and in part by the analysis of focus group findings.
1	Page 10, Line 22 – Please clarify whether the diagnosis within the last year is a recruitment criterion or not? Do you differentiate between acute and chronic patients in the analysis? Is that important for you?	Thank you for the questions. We have refined the description of eligibility criteria to clarify the point: participants must have new aphasia resulting from a new stroke.	Lines 209-210 People with aphasia (n=200) who meet the following eligibility criteria will be invited to participate: 1) admitted to a participating health service site; 2) new onset of aphasia caused by a new acute stroke, 3) aged ≥18 years; and 4) has capacity to consent by self or with a proxy. Lines 212-213 Patients admitted to participating health service sites with a new acute stroke will be screened for eligibility by clinical speech pathologists.
1	Page 10, Line 26-27 - Providing Information for Studies 1, 2, and 4: Ensure that information is consistent for all studies, and it may be necessary to provide information for all studies conducted as part of the	Thank you. In response to the reviewer’s comment, we have altered the manuscript to improve clarity on this point.	Lines 214-215 Eligible participants will be provided with written, aphasia-friendly participant information and consent forms describing the project, its aims, and the research activities. Speech pathologists will provide communication support throughout the consenting process. Lines 270-273

	project, also study 3?		When consenting for Study 1, participants will be provided with written, aphasia-friendly participant information and consent forms describing the research activities associated with Study 2. Speech pathologists will provide communication support throughout the consenting process. Lines 364-365 All prospective participants will be provided with communicatively accessible information about the study. A meeting will be held to answer questions and obtain demographic details for sampling. Participants who respond correctly to yes/no questions about the study's purpose and participation requirements will be invited to consent. Project investigators who are speech pathologists will provide communication support throughout the consenting process. Lines 389-392 When consenting for Study 1, participants will be provided with written, aphasia-friendly participant information and consent forms describing the research activities associated with Study 4. Speech pathologists will provide communication support throughout the consenting process.
1	Page 10, Line 37 – Have you discussed further ways to obtain informed consent? Especially for non-responsive patients. If people can't provide informed consent or are at the moment of recruitment non-responsive a proxy	This is described in Study 1 (p.8-9) and is re-iterated in Study 2 (p.10). We have added some text to clarify which sites the consent via proxy process applies to.	Lines 223-229 Consent via proxy: If a prospective participant agrees to participate but is not able to demonstrate capacity to provide informed consent at the time of recruitment, a proxy will be invited to consent on their behalf (Queensland and Victorian sites only). In such cases, treating clinicians will continue to monitor the participants' recovery and will provide further opportunities for the participant to provide their own consent. Patients admitted to Western Australian sites who are unable to demonstrate capacity to

	or legal representative might be invited to consent.		consent will be ineligible to participate, in accordance with Western Australian Department of Health legislation.
1	Page 10, Line 44 - Patient Interviews: Clarify whether speech pathologists will interview the patients or if patients are expected to complete the questionnaire themselves (online/paper).	Thank you for this comment. The manuscript has been updated to clarify these queries.	Lines 273-274 The survey will be in an aphasia-friendly format and administered in-person by speech pathology clinicians with provision of communication support.
1	Page 10-11 - Planned Analyses: Consider categorizing outcomes into primary and secondary outcomes for clarity in your analysis plan.	Thanks for this suggestion. On consideration, we determined it would be imprecise to present primary and secondary outcomes for this pilot work. The overarching objective is to gain understanding of the feasibility and acceptability of implementing the minimum dataset, which will be assessed using a range of methodologies.	No change.
1	Page 11, Line 16 - Missing Data: Since handling missing data is only need in time series, how many time points do you have? I can't find any information on how often the questionnaire is answered? What is the main observation duration? 1 week or 1 month after initial stroke?	Thank you for highlighting these points. The handling of missing data described in Study 2 relates to the completeness of the MEASuRES minimum dataset. We have relocated this information to more appropriately sit within Study 1. We have also revised the description of data collection, management, and	Lines 241-244 Data quality and completeness will be summarised using descriptive statistics. For each variable in the minimum dataset, the proportion completed will be calculated to determine patterns of missing, incomplete, or invalid data. Missing data will be handled in accordance with the processes outlined in existing guidelines (55, reference added). Lines 296-298 A log of site-training and factors

	Please define.	planned analyses in Studies 1 and 2 to improve the clarity of these sections. Additionally, we have also clarified the frequency and timing of completion of the surveys. The patient surveys will be completed prior to discharge from hospital. Length of hospital admission will vary, therefore the time post initial stroke will be variable across patients.	influencing implementation of the minimum dataset will be maintained, along with routinely collected administrative data. Lines 310-311 Findings from the different data sources (surveys, interviews, REDCap, training and implementation logs) will be integrated and compared using the Medical Research Council process evaluation framework (56), to understand the relationships between implementation, contextual factors, and other mechanisms of impact. Lines 268-270 (existing text reproduced here for context) All people with aphasia (or their proxy) who consented to participate in Study 1 (maximum n=200) will be invited to complete a survey prior to discharge from hospital to explore their experiences of completing the minimum dataset. Line 277 All speech pathologists who administer the minimum dataset data in Study 1 (n= ~25) will be invited to complete two online surveys throughout (timepoint 1) and at the completion (timepoint 2) of Study 1, to explore the performance of the minimum dataset, and their perspectives of administering the minimum dataset in clinical settings. In addition, expressions of interest will be sought from speech pathologists to participate in semi-structured interviews at the completion of Study 1.
1	Page 12, Line 52 - Scale Balance: Ensure the scale is balanced in both directions to prevent bias, and	The following details have been added to the development of the analysis tool section. As complete recovery	Lines 320-325 As a patient-rated measure, labels for the indicators on the anchor scale were selected based on ease of understanding by patients with aphasia. While an

	reevaluate the necessity of the "completely recovered" category.	from aphasia within the first six months highly likely (half of the initially affected stroke sufferers will completely recover their language abilities), we feel this is an important indicator to include on our anchor rating scale. Furthermore, research comparing balanced and unbalanced rating scales indicated that respondents rely predominantly on the meaning of the descriptors used to label the categories and less on the position of the response category (Friedman et al., 1981), so we do not expect a bias from using an inbalanced rating scale.	indicator for +3 (completely recovered) represents a potential perceived outcome for participants early after stroke onset, an acceptable indicator for -3 (a point lower than '-2 = much worse) could not be identified. As such, the anchor scale was developed with six response categories including a neutral response option, in line with existing research (reference added).
1	Page 13, Line 15-30 - Well described. Patients and Proxy Involvement: Include information on whether participants received incentives, the type, and the amount (e.g., monetary compensation).	Thanks. Details added.	Lines 366-372 To reduce the burden of participation, honorarium payments will be offered for time spent participating in Study 3. Participants with aphasia will be remunerated at a rate of 40 Australian Dollars (AUD) per hour, for an estimated total of five hours per participant (consent meeting, focus group, consensus workshop), in accordance with recommendations from Health Consumers Queensland (reference added). Speech Pathologists will be reimbursed at a rate of 60 AUD per hour for an estimated total of four hours per participant (focus group, consensus workshop), commensurate with the Queensland Health HP4 casual hourly rate.
2	What is the specific plan for acquisition of longitudinal data (i.e., will the same speech pathologists from the acute setting reach out to these patients and	Thank you for raising these questions. The following information has now been added to clarify these points.	Lines 167-169 These data will be collected for 200 participants with aphasia throughout the continuum of care across participating health service sites between January and August 2024.

	schedule follow-up research appointments? Will these be conducted in person or remotely? With what frequency will these evaluations be acquired? What are the preferred assessment time points beyond > 2 weeks post-stroke? etc.)		Lines 232-233 The continuity of speech pathologists collecting data will be dependent on usual care practices and service structure at each health service site. Lines 395-398 The second ROMA COS and anchor scales will be collected prior to discharge from inpatient care at the participating health service site. In instances where this is not feasible, outpatient follow up will be undertaken, following a flexible approach to delivery mode and assessor in line with site-specific agreements.
2	While this will be investigated in Study 2, the feasibility of acquiring the full MEASuRES dataset multiple times within the tumultuous period immediately following stroke may be reasonably expected to be somewhat limited. Do the authors have a hypothesis of how likely this is to be achievable, and/or a backup assessment plan if it proves to be too difficult to achieve? This may warrant addition to the limitations section.	Thank you for drawing our attention to some potential limitations of this work. Additions have been made as suggested.	Lines 174-177 Together, the four studies will provide preliminary information about the acceptability and feasibility of using the MEASuRES minimum dataset in routine clinical care. Embedded processes will allow exploration of where, when, and why components of the minimum dataset are not or cannot be collected to inform future refinement and scaling of the minimum dataset. Lines 236-237 (existing text reproduced here for context) In instances where elements of the minimum dataset are unable to be collected, reasons for missing or incomplete data will be recorded.
2	The plan for disentangling recovery due to time / neuroplasticity compared to recovery due to	Thank you for this comment. The primary aim of this project is an exploratory pilot of the acceptability and feasibility of administering the	Lines 38-39 Future studies employing the MEASuRES minimum dataset could integrate behavioural and imaging data to gain a fuller understanding of factors influencing aphasia recovery.

	treatment could be made clearer; is including time post stroke in regression models sufficient to address this?	MEASuRES minimum dataset in clinical care. Patient outcomes will be analysed alongside patient and clinician perceptions of change to develop minimal important change values. We will do some preliminary analyses to explore associations between patient characteristics, adherence to quality indicators and outcomes, however it is beyond the scope of the pilot study to complete complex analyses exploring all factors influencing recovery. We have further emphasised the primary aim of the study and have acknowledged this limitation.	Lines 173-174 Together, the four studies will provide preliminary information about the acceptability and feasibility of using the MEASuRES minimum dataset in routine clinical care. Lines 324-327 (existing text reproduced here for context) In study 4, the agreed thresholds on the anchor scale (Study 3) will be applied to treatment outcome data (Study 1), establishing MIC values for the ROMA COS.
2	Do the authors have a policy for or against incorporating neurological variables such as lesion size / lesion location into their models? These factors are likely to play a large role in the extent of recovery, and additionally may be confounded with treatment received. This may make interpretation of the specific influence of speech and language therapy in recovery more	Thank you for drawing our attention to the important role of neurological variables. Brain imaging is not part of the current MEASURES minimum dataset and this observational study without a control/untreated condition is not designed to disentangle spontaneous recovery and treatment effects. We have refined the strengths and limitations section to highlight the importance of this potential future direction once we have piloted and refined the dataset, to gain a fuller understanding of the	Lines 38-39 Future studies employing the MEASuRES minimum dataset could integrate behavioural and imaging data to gain a fuller understanding of factors influencing aphasia recovery.

	difficult. Do the authors have thoughts on the extent to which this is an issue for their purposes?	factors influencing recovery.	
2	How will the authors determine what speech language treatment was received by each patient? Is the assumption that this will always occur within the same hospital system that acquired the MEASuRES dataset and therefore be easily trackable?	Thank you for these questions. The speech language treatment received by each patient will be determined by the information collected in the Treatment Descriptors component of the MEASuRES minimum dataset. These treatment descriptors are outlined broadly in the background section. We have added further detail in Table 1. We have also reiterated the collection of the MEASuRES minimum dataset across the continuum of care at participating health services.	Lines 118-123 (existing text reproduced here for context) The International Population Registry for Aphasia after Stroke (30) includes a protocol for the documentation of aphasia treatment delivery. Information regarding treatment types, methods of delivery, and treatment dose are required to inform an understanding of post-stroke aphasia treatments (12, 37, 38). These treatment descriptors would support the identification of outcome predictors (12), however, they are not routinely collected. Table 1 inserted in the Introduction section (line 97). Lines 167-169 These data will be collected for 200 participants with aphasia throughout the continuum of care across participating health service sites between January and August 2024.
2	How were the desired sample sizes (e.g. 200 people with aphasia in Study 1, 10 each people with aphasia and speech pathologists in Study 3) chosen?	Thank you for raising these questions. Information has been added to clarify these points.	Lines 169-171 These data will be collected for 200 participants with aphasia throughout the continuum of care across participating health service sites between January and August 2024. The sample size was informed by recommendations for conducting and reporting MIC studies [47], aphasia admission rates at each of the participating sites, and power calculations for sub-group analyses for Studies 1 and 4. Lines 284-287 Purposive sampling of clinicians across

			years of experience, clinical setting, and health service sites will be applied to ensure a range of viewpoints. These interviews will serve to contextualise survey results and provide further insights from the participating clinicians. Additional interviews will be conducted where required. Lines 351-356 Participants will be consented to the study in alignment with the sampling criteria. It is anticipated that 10 people with post-stroke aphasia and 10 speech pathologists will be sufficient to inform a threshold for meaningful change . Each participant will attend one focus group. A (maximum of five participants will attend each per group.). The final number of focus groups will be determined in part by the amount of time, and level of communication assistance, each participant will require to understand and meaningfully participate, and in part by the analysis of focus group findings.
--	--	--	---

VERSION 2 – REVIEW

REVIEWER	Daniel Schindel Charité Universitätsmedizin Berlin, Institute of Medical Sociology and Rehabilitation Science
REVIEW RETURNED	13-Feb-2024

GENERAL COMMENTS	Dear authors, Thank you for the detailed revision. Two closing thoughts. (1) It would be useful to have more information on how the items of the MEASuRES minimum dataset are scored. Please consider providing more details (on scales or response options). If the questionnaire is not yet finalised, please consider publishing it together with the next publication on the results of the pilot project. (2) Both lines 92 and 117: Since you introduce an abbreviation with capital letters, please add the acronym of the study, i.e. I-PRAISE. I wish you all the best for the further realisation of your studies.
---

REVIEWER	Deborah Levy University of California San Francisco, Neurosurgery
REVIEW RETURNED	06-Feb-2024

GENERAL COMMENTS	The authors have sufficiently addressed my concerns. I look forward to hearing about the results of their study!
--